# Cranial Investigations of Crested Porcupine (*Hystrix cristata*) by Anatomical Cross-Sections and Magnetic Resonance Imaging

**DOI:** 10.3390/ani13162551

**Published:** 2023-08-08

**Authors:** Daniel Morales-Bordon, Mario Encinoso, Alberto Arencibia, José Raduan Jaber

**Affiliations:** 1Departamento de Patología Animal, Producción Animal, Bromatología y Tecnología de los Alimentos, Facultad de Veterinaria, Universidad de Las Palmas de Gran Canaria, Trasmontaña, 35413 Arucas, Las Palmas, Spain; daniel.morales@ulpgc.es; 2Hospital Clínico Veterinario, Facultad de Veterinaria, Universidad de Las Palmas de Gran Canaria, Trasmontaña, 35413 Arucas, Las Palmas, Spain; 3Departamento de Morfologia, Facultad de Veterinaria, Universidad de Las Palmas de Gran Canaria, Trasmontaña, 35413 Arucas, Las Palmas, Spain; alberto.arencibia@ulpgc.es

**Keywords:** magnetic resonance imaging, anatomical sections, rodents, head anatomy, CNS, crested porcupine

## Abstract

**Simple Summary:**

Advanced imaging diagnostic techniques, such as magnetic resonance imaging with anatomical sections, were used to evaluate the head of the crested porcupine (*Hystrix cristata*). These techniques were very helpful to delineate the main formations that compose the central nervous system (CNS), as well as its associated structures. To the authors’ knowledge, the present study is the first to describe this area using anatomical sections and magnetic resonance imaging (MRI) in crested porcupines.

**Abstract:**

This paper aimed to describe an atlas of the crested porcupine (*Hystrix cristata*) head by applying advanced imaging techniques such as MRI. Furthermore, by combining the images acquired through these techniques with anatomical sections, we obtained an adequate description of the structures that form the CNS and associated structures of this species. This anatomical information could serve as a valuable diagnostic tool for the clinical evaluation of different pathological processes in porcupines, such as abscesses, skull malformations, fractures, and neoplasia.

## 1. Introduction

In recent years, advanced diagnostic imaging techniques have facilitated the visualization of several diseases in captive and wildlife medicine. Traditionally, standard radiography was the choice more frequently used by clinicians, biologists, and researchers [1]. Nevertheless, the results of previous studies have demonstrated that computed tomography (CT) and magnetic resonance imaging (MRI) provide more information with which to improve diagnostic accuracy, prognosis, treatment of diseases, and anatomic knowledge [2,3,4]. These modern techniques avoid the superimposition of adjacent anatomical structures and delineate the anatomic detail of specific tissue densities more finely, which improving the capacity for interpretation [2]. These benefits provide important value for the anatomic investigations of specific regions, providing essential knowledge of domestic and exotic mammal species [2,3,4,5]. Among these species, we highlight the crested porcupine (*Hystrix cristata*), which is one of the most well-known members of the family Hystricidae and is included in the IUCN red list as least concern since they are regionally or locally threatened, mainly due to the fact that they are hunted illegally for food and killed because they are considered as an agricultural pest [6,7]. Consequently, these animals require appropriate conservation policies in regional and local contexts. It is a species of rodent native to Italy and Sicily, as well as a broad central strip ranging from Senegal west to Somalia and east to Kenya and Tanzania, but very little is known regarding its geographic variability in Africa [6,7,8]. These animals are found in forests, rocky areas, mountains, croplands, and sandhill deserts, and protect themselves in caves, aardvark holes, and burrows that they dig themselves [6,7].

Porcupines are characterized for presenting strong and pointed quills covering the tail, sides, and top of the body that can be raised into a crest and used for defensive purposes. Concerning their heads, they are large and robust with enlarged infraorbital foramina so that portions of the masseter extend through it and arise from the frontal side surface of the snout. In addition, the nasal cavities are enlarged [9]. Interestingly, Hystrix show prominent pockets in the skull, maxillary, lacrimal, and turbinate bones, which could be used for attachment of the masticatory muscles.

Some valuable literature on the anatomical, physiological, and pathological study of pet rabbits and rodents is already broadly available [10,11,12,13,14,15,16]. Among these, we highlighted those studies that deal with brain and head MRI anatomy, which are available at reduced resolution from low field-strength and anatomical cross-sections, and more recently, with a high field-strength magnet in excised and fixed rabbit brain for research and clinical guidance in animals with diverse pathological processes [14,15]. To the best of the authors’ knowledge, only a few publications have been conducted on porcupines, which have been focused on their biology, morphometric geographic variability, and some clinical conditions such as upper respiratory tract disease [6,8,9,13]. However, no description of the brain has been reported in this specific rodent. Therefore, this study aimed to describe the normal anatomy of the CNS of the crested porcupine and its associated structures, using specific anatomical sections that better matched the images obtained by MRI. The combination of MRI and macroscopic anatomical sections could provide helpful information for anatomic teaching and clinical practice.

## 2. Materials and Methods

### 2.1. Animals

Three carcasses of adult (two males and one female) crested porcupines (*Hystrix cristata)* from the zoological park “Rancho Texas Lanzarote Park” (Lanzarote, Canary Islands, Spain) were collected. None of the porcupines used in this study had a history of central nervous system disease, and no abnormalities were detected during imaging examinations. Prior consent was obtained from the responsible person at the zoological park to include the porcupines in this study.

### 2.2. Anatomic Evaluation

After the MR images were acquired, the two scanned carcasses were frozen at −80 °C for 48 h before performing anatomical cross-sections. After that, the two frozen carcasses were sectioned using an electric band saw to obtain sequential anatomical cross-sections. Contiguous 1 cm transverse slices were obtained starting at the olfactory bulb and extending to the first cervical vertebra region. These slices were intentionally thicker than those of the MRI to retain the integrity and position of the anatomic formations in the slices. The obtained sections were cleaned with water, numbered, and photographed on the cranial and caudal surfaces. Later, we selected those anatomical sections that better correlated with the MRI images to identify the structures of central nervous system and associated structures of the crested porcupine. Moreover, we also used anatomical textbooks and relevant references describing the anatomy of cats, rabbits, and different rodent species to facilitate accurate anatomic identification of these structures [14,15,17,18,19,20,21].

### 2.3. MRI Technique

A magnetic resonance imaging study was conducted on individual crested porcupine using a 1.5-Tesla magnet (Toshiba, Vantage Elan, Japan) while the animal was positioned in a ventral recumbent position. A standard MRI protocol was employed to generate spin-echo (SE) T1-weighted and T2-weighted images in sagittal, transverse, and dorsal planes. SE T1-weighted transverse images were acquired with the following parameters: echo time (TE) of 10 ms, repetition time (TR) of 800 ms, acquisition matrix size of 536 × 384, and a slice thickness of 4.5 mm with 4 mm spacing between slices. For SE T2-weighted transverse images, the TE was set to 120 ms, TR to 10,541 ms, acquisition matrix size to 624 × 448, and slice thickness to 3 mm with 3 mm interslice spacing. SE T2-weighted sagittal images were obtained with a TE of 120 ms, TR of 7529 ms, acquisition matrix size of 512 × 804, and a slice thickness of 2.8 mm with 2 mm interslice spacing. Lastly, SE T2-weighted dorsal images were acquired using a TE of 120 ms, TR of 8282 ms, acquisition matrix size of 468 × 512, and a slice thickness of 3.4 mm with 3 mm interslice spacing. A medical imaging viewer (OsiriX MD, Geneva, Switzerland) was utilized to assess the images of the study.

## 3. Results

In this study, we present different anatomical cross-sections and T2W images of the crested porcupine, which were displayed in a rostral-to-caudal progression from the level of the ethmoturbinates and eyeballs to the caudal end of the brain stem. Therefore, Figure 1 represents a T2W sagittal image in which each line and number (I–VI) represents approximately the level of the following anatomical and MRI transverse images. Transverse anatomical sections and MR images of the porcupine revealing the relevant anatomical structures of the head were displayed (Figure 2, Figure 3, Figure 4, Figure 5, Figure 6 and Figure 7). These figures were composed of two images: (A) macroscopic anatomical section and (B) T2W MR image. Additionally, two sagittal and two dorsal T2W MR images were presented to depict the relevant structures of the porcupine central nervous system (Figure 8 and Figure 9).

### 3.1. Anatomical Cross-Sections

The anatomical sections obtained in this study allowed us to visualize the different structures belonging to the central nervous system and its associated structures, which were labeled according to the International Committee on Veterinary Gross Anatomical Nomenclature. Therefore, we identified the main components of the brain (the prosencephalon, mesencephalon, and rhombencephalon). Thus, the two telencephalic hemispheres surrounded by the cerebral cortex and separated by the longitudinal cerebral fissure were identified (Figure 5A, Figure 6A and Figure 7A). Both hemispheres were connected by fibers of white matter known as the corpus callosum (Figure 4A, Figure 5A, Figure 6A and Figure 7A). Each cerebral hemisphere contained a lateral ventricle (Figure 4A, Figure 5A, Figure 6A and Figure 7A). Ventrally, we distinguished a component of the basal ganglia, the nucleus caudatus (Figure 4A and Figure 5A). Thus, we identified different parts of it, such as the head and the tail. Other structures which we observed were the septal nuclei, which were circumscribed by two parallel vertical lines through the most inferior and medial aspect of each lateral ventricle (Figure 4A). More caudally, the diencephalon enclosing the third ventricle was identified, as well as more ventrally specific components of the hypothalamus, such as the optic chiasm (Figure 4A and Figure 5A). Additionally, these sections were quite helpful in showing the caudal parts of the thalamus. Hence, the lateral eminence on the caudodorsal surface of the thalamus, known as the lateral geniculate body, was distinguished, whereas caudoventrally, we identified the medial geniculate body of the thalamus (Figure 5A). Moreover, the dorsal part of the mesencephalon with the caudal and rostral colliculus and its ventral part with the cerebral peduncles were also shown (Figure 6A and Figure 7A). These anatomical sections were helpful in identifying the vermis and the cerebellar peduncles, which connected the cerebellum to the adjacent brain stem and the cerebrum (Figure 7A). The ventral part of the cerebellum with the lingula, covering part of the fourth ventricle, could also be identified (Figure 7A). These sections were also helpful in distinguishing the medulla oblongata and the decussation of the pyramids. In addition, different bony structures comprising the neurocranium were observed, such as the frontal, the temporal (with its scamous, petrous, and tympanic parts), the sphenoid, and the occipital bones (Figure 2A, Figure 3A, Figure 4A, Figure 5A, Figure 6A and Figure 7A). Furthermore, these sections showed different air-filled spaces, such as the frontal and the sphenoidal sinuses (Figure 2A, Figure 3A, Figure 4A, Figure 5A, Figure 6A and Figure 7A), and structures associated with the nasal cavity, including the ethmoturbinates and the vomer, could be distinguished (Figure 2A and Figure 3A). Main sensory organs such as the eyeball and its associated structures were also depicted. Consequently, we identified the retina, the vitreous chamber, and the optic nerve, which were surrounded by extraocular muscles. Among these, we distinguished the dorsal and ventral rectus muscles (Figure 2A). Moreover, the main components of the auditory system, such as the external auditory canal, the tympanic cavity, and the inner ear, were visualized (Figure 5A and Figure 6A). Also, we identified the relevant muscles related to masticatory function, such as the temporalis, the medial and lateral pterygoid muscles, the masseter, the digastric muscle, and other important muscles, including the buccinator muscle and the longissimus capitis of the head (Figure 2A, Figure 3A, Figure 4A, Figure 5A, Figure 6A and Figure 7A).

### 3.2. Magnetic Resonance Imaging (MRI)

No significant anatomic differences were identified subjectively in the three porcupines which were imaged. Most anatomic structures distinguished on T2-weighted images of the cadaver specimens matched adequately with structures identified in the corresponding anatomical cross-sections. Hence, the central nervous system structures of the porcupine head, the eyeball’s structures (vitreous humour and lens), and the masticatory muscles showed an accurate visualization using T2W MR images. Nonetheless, the bones that comprised the neurocranium, such as the frontal, the parietal, the temporal, the occipital, and the sphenoid bones, were identified with a hypointense signal (Figure 2B, Figure 3B, Figure 4B, Figure 5B, Figure 6B, Figure 7B, Figure 8 and Figure 9).

In the transverse planes of the encephalon, identifiable structures of the brain were more hyperintense than the white matter, which was more hypointense in T2W sequences (Figure 3B, Figure 4B, Figure 5B, Figure 6B and Figure 7B). Moreover, the two sagittal and dorsal (Figure 8 and Figure 9) images and the different transverse T2W images were essential to depicting the components that comprise the ventricular system, which displayed a hyperintense signal. Hence, the lateral ventricles and the dorsal and ventral parts of the third ventricle were displayed (Figure 4B, Figure 5B, Figure 6B, Figure 7B, Figure 8 and Figure 9), and among these, we observed interthalamic adhesion, limited laterally by the right and left sides of the thalamus (Figure 8A). In addition, transverse, sagittal, and parasagittal T2W images displayed with adequate detail the dorsal and ventral parts of the hippocampus (Figure 5B, Figure 6B, Figure 7B, Figure 8B and Figure 9). In addition, the tectum of mesencephalon (tectum mesencephali) with the caudal colliculus and the fourth ventricle was visualized in excellent detail (Figure 8A,B and Figure 9B). Other essential components of the CNS, such as the vermis of the cerebellum with its dorsal and ventral lobes, were distinguished in the sagittal T2W images (Figure 8A). Hence, the moderate contrast between grey and white matter was helpful in distinguishing the different lobes. Thus, the dorsal lobes of the cerebellum (the rostral and ventral culmen, the declive, the folium, the tuber, and the pyramid), as well as its ventral lobes (the lingula, the nodulus, and the uvula), were identified. In addition to these findings, we also observed the rostral and caudal cerebellar peduncles (Figure 7B and Figure 9A). This technique also facilitated an adequate resolution to be achieved with which to identify the muscles involved in the masticatory function, which have already been mentioned in the anatomical sections.

## 4. Discussion

In recent years, modern diagnostic imaging techniques such as computed tomography and magnetic resonance imaging have become quite fashionable in captive and free-ranging animals to diagnose and treat diseases, since their availability for clinical use has dramatically increased. However, the costs and risk of complications due to general anesthesia [11,12] are limiting factors to the use of these techniques in exotic animals. In contrast with traditional imaging methods such as radiography and ultrasound, which are widespread among veterinarians [10], these procedures can provide images of the different structures in various planes without repositioning the animal [12,16,22,23]. CT provides more detailed anatomical information regarding the skull bones and dentition compared to MRI [16]. However, certain clinical conditions affecting rodents may involve secondary soft tissue structures that are better visualized using MRI. In the case of the CNS, magnetic resonance imaging is considered the gold standard imaging modality for humans and animals, as CT tends to depict the CNS structures as a homogeneous formation without a clear distinction between its different components [24]. Furthermore, MRI allows the identification of structures deep into the bone and air and is less limited by operator experience [3,11,12,14]. Therefore, these advanced imaging techniques enhance anatomic identification and lesion detection, allowing assessment, detailed prognosis, diagnosis of underlying lesions, and treatment choice to be carried out [11,12]. In the case of rodent species, these procedures have been used to evaluate a range of head processes not well visualized through conventional imaging techniques. Among these, we included dental disease and its associated problems, deformities, and osteomyelitis; the extension of the infection process to different bone cavities of the skull such as the nasal cavity, the paranasal sinuses, and the tympanic bullae; as well as fractures and different types of neoplasia [10,11,12]. However, it is important to consider their cost, the need for sedation, and the time required for data analysis before employing these methods. Despite these disadvantages, their application in the study of endangered species is justified due to the valuable information they can provide with minimal risk to the animal [3,22,23]. In this study, we successfully yielded high-resolution images of the central nervous system and its related structures using state-of-the-art diagnostic imaging techniques such as MRI. These images exhibited exceptional clarity and complemented the anatomical cross-section images. To the best of the authors’ knowledge, this represents the first comprehensive description of the CNS and associated structures of the crested porcupine utilizing contemporary diagnostic imaging techniques, such as MRI, and their correlation with the anatomical cross-section images. As a result, our investigation has provided valuable insights into the intricate anatomic details of the crested porcupine’s head, which hold potential diagnostic, experimental, and educational significance.

In our study, the utilization of anatomical cross-sections has proven to be immensely valuable in accurately characterizing the morphologic features of the CNS in the crested porcupine, as well as the structures related to the eyeball and the auditory system. These cross-sectional views played a crucial role in visualizing significant components of the diencephalon, brainstem, and cerebellum, including the habenula, the rostral and caudal colliculi, and the cerebellar peduncles. However, due to the wide interval between slices employed in this study, some of these structures were not clearly discernible in the transverse MR views. Comparable investigations conducted in other exotic species, such as rabbits, guinea pigs, iguanas, and loggerhead turtles, have consistently demonstrated that this combination is essential to comparing the relative positions and sizes of cephalic or other anatomical structures [3,14,25,26,27].

The MR images obtained in the transverse and sagittal planes were acquired without repositioning the head. The high-field magnet facilitated an adequate evaluation of the crested porcupine’s head. Thus, the different planes utilized herein made structure identification more evident. Additionally, subjective image analysis and objective measurements revealed a relatively large dorsal *metencephalon* and a smaller *telencephalon* compared to dogs of a similar size and weight. One report described similar findings in size when they studied the rabbit brain with a high field-strength magnet [15]. Nonetheless, further studies are needed to confirm these findings. Moreover, the present study demonstrated that MRI was effective in visualizing the bones of the porcupine’s head. Nonetheless, it is important to note, as described in other reports, that the skull bones and teeth could be identified due to their hypointense signal [24,27]. In contrast, excellent discrimination of the main components of the *encephalon*, including the *rhinencephalon*, *telencephalon*, *diencephalon*, *mesencephalon*, *metencephalon*, and *myelencephalon*, was achieved with the T2W images. These images have proven to be useful for anatomical and clinical studies of various exotic species [3,12,22,23,24,25,26,27,28,29,30,31]. Therefore, the transverse T2W images revealed the olfactory bulb and its recess, which was hyperattenuated compared to the *telencephalon*. Similar enhancement was observed in rabbits, particularly in the sagittal plane [14,15]. Additionally, the sagittal T2W images were essential for visualizing the different lobes of the cerebellar vermis, which displayed a hyperintense/isointense signal when compared with the hypointense cerebellar white matter. Our results confirmed those obtained in rodents and rabbits, which have demonstrated that MRI is an essential technique for evaluating the CNS and its associated soft tissue structures [2,11,12,14,15,16,17,27,31,32]. However, other structures, such as the nasal cavity and paranasal sinuses, were not evaluated in our study because we were focused on those structures better visualized by MRI.

As mentioned in previous studies [25], the use of cadavers in our research limited the administration of an intravenous contrast medium, which could have improved the resolution of MR images. This limitation, as well as the small number of specimens, should be considered in further studies. Nevertheless, both MRI and macroscopic anatomical sections provided adequate information for anatomic evaluation in teaching and clinical settings. In contrast to other studies conducted on rodents and rabbits [15,30,31,32], we employed anatomical sections and conducted a comprehensive anatomical description, including the location and intensity of the different structures comprising the central nervous system of the crested porcupine. The sagittal images, compared with those displayed in the transverse plane, facilitated a better assessment of the topographic anatomical structures in the median plane, primarily involving the intracranial cavity and the central nervous system. Similar findings have been reported in other MRI studies performed on exotic species [3,25]. All the images obtained in this study could serve as initial reference materials to support pathological investigations of crested porcupine heads.

## 5. Conclusions

This investigation is the first description of a crested porcupine’s head using transverse, sagittal, and dorsal MR images in combination with anatomical cross-sections. The images obtained in this study were quite helpful in providing essential references for the different bone and soft tissue structures comprising the CNS and sensory organs of the crested porcupine. Therefore, the information obtained in this study could be adequate for the anatomic and clinical evaluation of numerous pathologic processes involving the heads of these animals, such as abscesses, metabolic bone diseases, fractures, inflammation, and neoplasia. Moreover, the MR images obtained in different spatial planes could facilitate our understanding of anatomic organization for our students, since these procedures allow for the visualization of structures without overlapping, eliminating the difficulties of visualizing specific anatomic structures. Nevertheless, the high cost and accessibility of this equipment do not facilitate its use on porcupines or rodents in daily veterinary practice.

## Figures and Tables

**Figure 1 animals-13-02551-f001:**
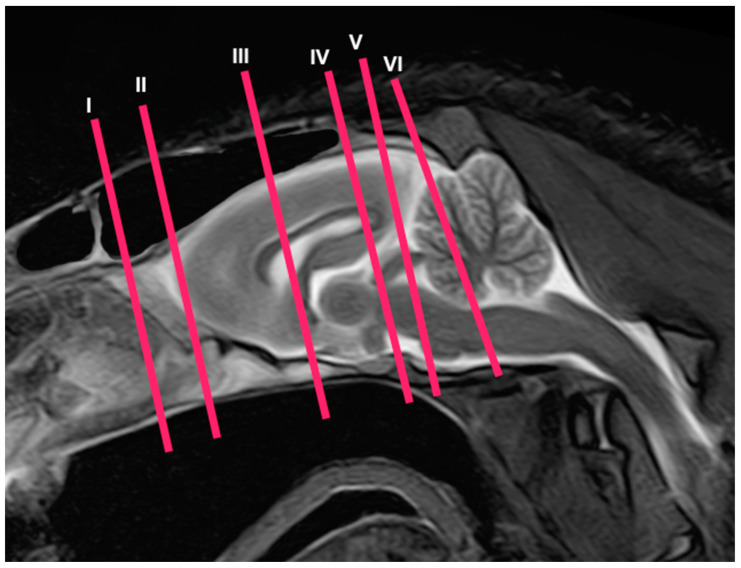
Sagittal T2W MR image of the head of a crested porcupine. The lines and numbers (I–VI) represent the approximate levels of the following transverse cross-sections and MR images.

**Figure 2 animals-13-02551-f002:**
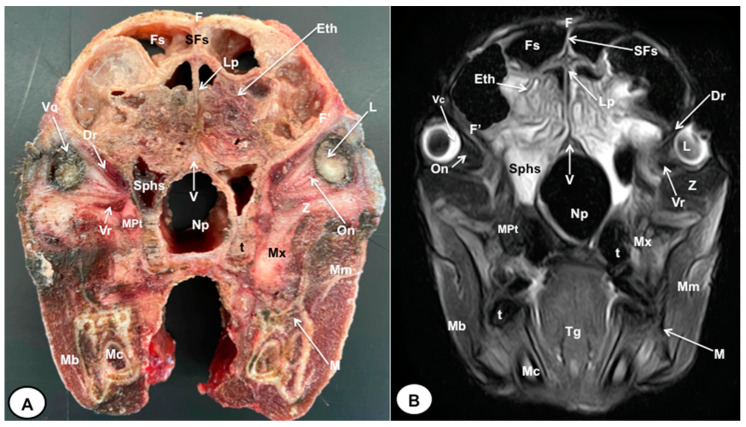
Transverse cross-section (**A**) and T2W MR (**B**) images of the crested porcupine head at the level of the ethmoturbinates corresponding to line I in Figure 1. F: frontal bone; F′: orbital plate of frontal bone; Fs: frontal sinuses; SFs: septum of frontal sinuses; Eth: ethmoturbinates; Lp: *Lamina perpendicularis ossis ethmoidei;* Sphs: sphenoid sinus; V: vomer; Np: nasopharynx; On: optic nerve; Vc: vitreous chamber; L: lens; Dr: *Musculus rectus dorsalis*; Vr: *Musculus rectus ventralis*; MPt: *Musculus pterygoideus medialis*; Z: zygomatic bone; Mx: maxillary bone; t: tooth; M: mandible; Mc: mandibular canal; Mm: *Musculus masseter*; Mb: *Musculus buccinator*; Tg: tongue.

**Figure 3 animals-13-02551-f003:**
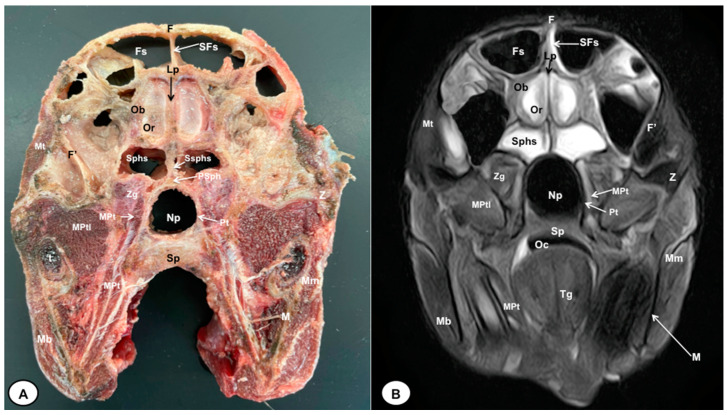
Transverse cross-section (**A**) and T2W MR (**B**) images of the crested porcupine head at the level of the olfactory bulb corresponding to line II in Figure 1. F: frontal bone; F′: orbital plate of frontal bone; Fs: frontal sinuses; SFs: septum of frontal sinuses; Ob: olfactory bulb; Or: olfactory recess; Lp: *Lamina perpendicularis ossis ethmoidei;* Sphs: sphenoid sinus; Ssphs: septum of sphenoidal sinuses; PSph: presphenoid bone; Np: nasopharynx; Pt: pterygoid bone; Sp: soft palate; Oc: oral cavity; MPt: *Musculus pterygoideus medialis*; MPtl: *Musculus pterygoideus lateralis*; Mt: *Musculus temporalis;* Zg: zygomatic glands; Z: zygomatic bone; t: tooth; M: mandible; Mm: *Musculus masseter*; Mb: *Musculus buccinator*; Tg: tongue.

**Figure 4 animals-13-02551-f004:**
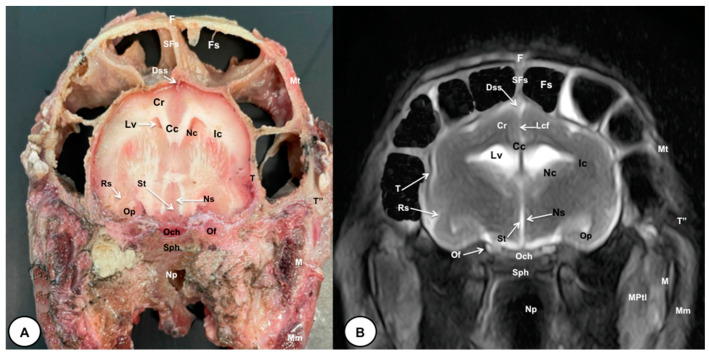
Transverse cross-section (**A**) and T2W MR (**B**) images of the crested porcupine head at the level of the optic chiasm, corresponding to line III in Figure 1. F: frontal bone; Fs: frontal sinuses; SFs: septum of frontal sinuses; Dss: dorsal sagittal sinus; Lcf: longitudinal cerebral fissure; Cc: corpus callosus; Cr: *Corona radiata*; Lv: lateral ventricle; Nc: *Nucleus caudatus;* Ic: internal capsule; Rs: rhinal sulcus; Op: olfactory peduncle; St: septum of telencephalon (*Septum telencephali*); Ns: *Nuclei septi*; Och: optic chiasm; Of: orbital fissure; T: temporal bone (squamous part); T″: zygomatic process of temporal bone; Sph: sphenoid bone; Np: nasopharynx; Mt: *Musculus temporalis*; MPtl: *Musculus pterygoideus lateralis*; M: mandible. Mm: *Musculus masseter*.

**Figure 5 animals-13-02551-f005:**
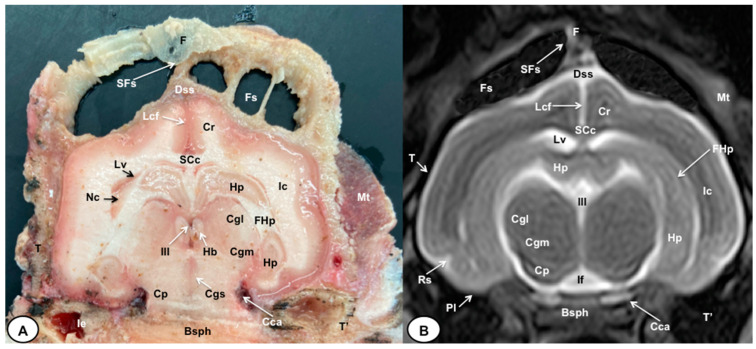
Transverse cross-section (**A**) and T2W MR (**B**) images of the crested porcupine head at the level of the caudal part of the diencephalon, corresponding to line IV in Figure 1. F: frontal bone; Fs: frontal sinuses; SFs: septum of frontal sinuses; Dss: dorsal sagittal sinus; Lcf: longitudinal cerebral fissure; SCc: splenium of corpus callosum; Cr: *Corona radiata*; Lv: lateral ventricle; Hp: *Hippocampus*; FHp. fimbria of hippocampus; Nc: *Nucleus caudatus* (tail); Ic: internal capsule; III: third ventricle; Hb: habenula; Cgl: *Corpus geniculatum laterale* (lateral geniculate body); Cgm: *Corpus geniculatum mediale* (medial geniculate body); Cgs: central grey substance; Cp: cerebral peduncle; If: interpeduncular fossa; Rs: rhinal sulcus; Pl: piriform lobe; T: temporal bone squama; T′: tympanic and petrous parts of temporal bone; Ie: inner ear. Cca: caudal communicating artery; Bsph: basisphenoid bone; Mt: *Musculus temporalis*.

**Figure 6 animals-13-02551-f006:**
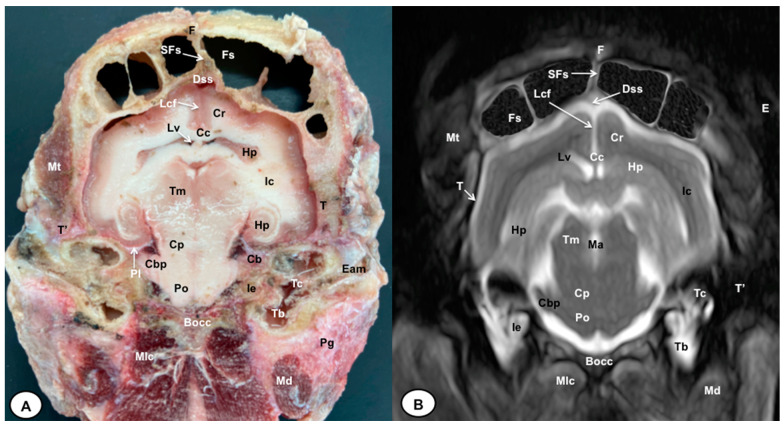
Transverse cross-section (**A**) and T2W MR (**B**) images of the crested porcupine head at the level of the caudal part of the hippocampus, corresponding to line V in Figure 1. F: frontal bone; Fs: frontal sinuses; SFs: septum of frontal sinuses; Dss: dorsal sagittal sinus; Lcf: longitudinal cerebral fissure; Cc: corpus callosus; Cr: *Corona radiata*; Lv: lateral ventricle; Hp: *Hippocampus*; Ic: internal capsule; Tm: tectum of mesencephalon; Ma: mesencephalic aqueduct; Po: Pons; Pl: piriform lobe; Cb: cerebellum; Cbp: cerebellar peduncle (lateral); Cp: cerebral peduncle; T: temporal bone squama; T′: tympanic and petrous parts of temporal bone; Bocc: basioccipital bone; Mt: *Musculus temporalis*; Mlc: *Musculus longissimus capitis*; Md: *Musculus digastricus*; Pg: parotid gland; Ie: inner ear (cochlea); Tc: tympanic cavity; Tb: tympanic bulla; E: ear (external part).

**Figure 7 animals-13-02551-f007:**
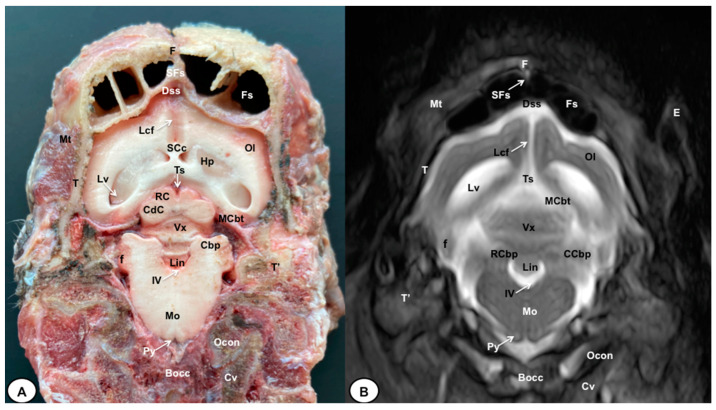
Transverse cross-section (**A**) and T2W MR (**B**) images of the crested porcupine head at the level of the cerebellum, corresponding to line VI in Figure 1. F: frontal bone; Fs: frontal sinuses; SFs: septum of frontal sinuses; T: temporal bone squama; Mt: *Musculus temporalis*; T′: tympanic and petrous parts of temporal bone; Dss: dorsal sagittal sinus; Lcf: longitudinal cerebral fissure; Lv: lateral ventricle; SCc: splenium of corpus callosum; Hp: hippocampus; Ts: transverse sinus; MCbt: membranous cerebellar tentorium; RC: rostral colliculus; CdC: caudal colliculus; Cbp: cerebellar peduncle; Vx: vermis of cerebellum; RCbp: rostral cerebellar peduncle; CCbp: caudal cerebellar peduncle; Lin: lingula; IV: fourth ventricle; Mo: *medulla oblongata*; Py: pyramids of the medulla oblongata; Cv: first cervical vertebra; Ocon: occipital condyle; Bocc: basioccipital bone; E: external ear.

**Figure 8 animals-13-02551-f008:**
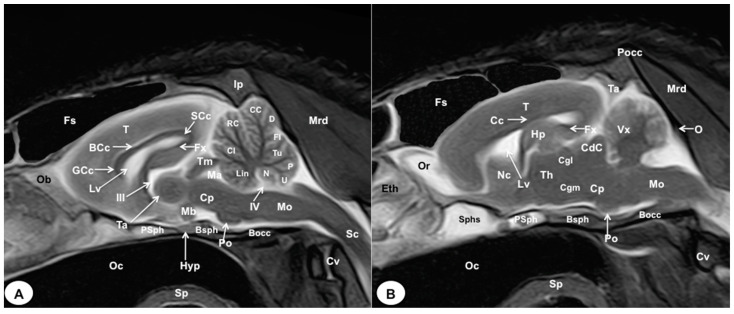
Sagittal (**A**) and parasagittal (**B**) T2W MR images of the crested porcupine head. Fs: frontal sinuses; Ip: interparietal bone; Pocc: *Protuberantia occipitalis externa;* Eth: ethmoturbinates; Ob: olfactory bulb; Or: olfactory recess; T: telencephalon; Lv: lateral ventricle; Cc: *corpus callosum*; GCc: genu of corpus callosum; BCc: body of corpus callosum; SCc: splenium of corpus callosum; Hp: *hippocampus*; Fx: fornix; Nc: *Nucleus caudatus*; III: third ventricle; Ta: interthalamic adhesion; Th: *thalamus;* Cgl: *Corpus geniculatum laterale* (lateral geniculate body); Cgm: *Corpus geniculatum mediale* (medial geniculate body); Mb: mamillary body; Hyp: *Hypophysis*; Ta: tentorial process; Tm: tectum of mesencephalon; Ma: mesencephalic aqueduct; CdC: caudal colliculus; Cp: cerebral peduncle; Vx: vermis of cerebellum; Lin: lingula; Cl: central lobe of cerebellum; RC: rostral culmen; CC: caudal culmen; D: declive; Fl: folium; Tu: tuber; P: pyramid; U: uvula; N: nodule; IV: fourth ventricle; Po: Pons; Mo: *Medulla oblongata*; Sc: spinal cord; Cv: cervical vertebra; Mrd: *Musculus rectus dorsalis;* O: occipital bone; Sphs: sphenoid sinus; PSph: presphenoid bone; Bsph: basisphenoid bone; Bocc: basioccipital bone; Oc: oral cavity; Sp: soft palate.

**Figure 9 animals-13-02551-f009:**
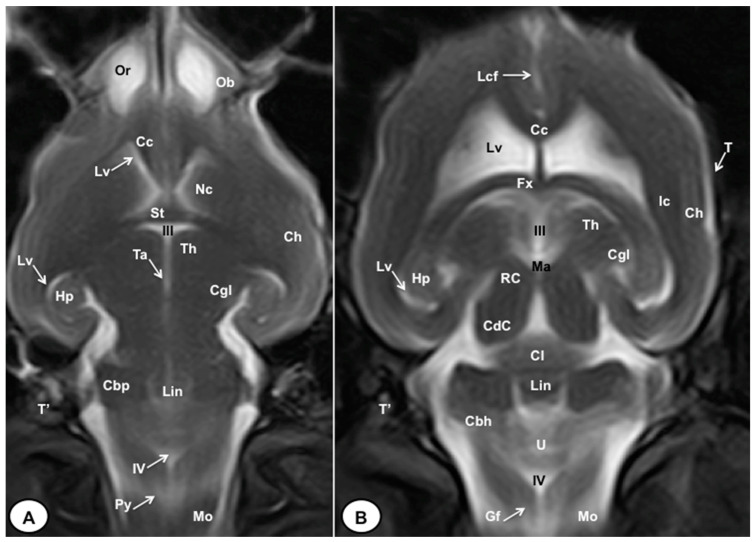
T2W dorsal MRI images of the crested porcupine head at the level of the olfactory bulb (**A**) and corpus callosum (**B**). Ob: olfactory bulb; Or: olfactory recess; Lcf: longitudinal cerebral fissure; Cc: corpus callosum; Lv: lateral ventricle; Nc: *Nucleus caudatus*; St: septum of telencephalon (*Septum telencephali)*; Fx: *fornix*; III: third ventricle; Ic: internal capsule; Ch: cerebral hemisphere; Hp: *hippocampus*; Th. *thalamus*; Ta: interthalamic adhesion; Cgl: *corpus geniculatum laterale* (lateral geniculate body); Ma: mesencephalic aqueduct; RC: rostral colliculus; CdC: caudal colliculus; Cl: central lobe of cerebellum; Lin: *lingula*; Cbp: cerebellar peduncle; Cbh: cerebellar hemisphere; IV: fourth ventricle; Mo: *medulla oblongata*; Py: pyramids of the medulla oblongata; Gf: Gracile fasciculus; T: temporal bone squama; T′: tympanic and petrous parts of temporal bone.

## Data Availability

The information is available at “https://accedacris.ulpgc.es”.

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
