# Peer review of "Cranial Investigations of Crested Porcupine (*Hystrix cristata*) by Anatomical Cross-Sections and Magnetic Resonance Imaging"

_animals, 2023, doi:10.3390/ani13162551_

Round 1
Reviewer 1 Report
The present study investigated the cranial structures of crested porcopines by CT and MRI
The study is well-oriented and novel but there are some points to be addressed.
Major points:
There is no ethical approval
Discussion is weak and lacks comparison with other animals espicially rodents with focus on paranasal sinuses
https://onlinelibrary.wiley.com/doi/full/10.1111/ahe.12752
https://pubmed.ncbi.nlm.nih.gov/19794893/
Minor points:
MRI in abstract should be defined.
Replace hour by h.
L91 Replace anatomical-gross sections by anatomical cross-sections
In the whole manuscript please replace gross-sections by anatomical sections
Results should be written in the paste tense
SNC appeared suddenly in conclusion?
The results section requires correction. Both paste and present tenses are used
Author Response
Dear Reviewer,
First, we appreciate the comments and suggestions, which have been quite useful in improving our manuscript.
Major points
- There is no ethical approval: Since we were working with cadavers, ethical approval was not necessary. Instead, we have provided informed consent explaining that the person in charge from Rancho Texas Lanzarote Park allowed us to perform the study.
- The discussion is weak and lacks comparison with other animals, especially rodents with a focus on paranasal sinuses: Following your recommendation, we have modified the discussion section, adding information compared to other animals after consulting the recommended references.
Minor points
- MRI in the abstract should be defined: the information has been defined.
- We have replaced hour with h.
- L91: we have replaced "anatomical-gross sections" with "anatomical cross-sections", as well as replaced "gross-sections" with "anatomical sections" along the manuscript.
- Results should be written in the past tense: As you suggested, results have been written in the past tense.
- SNC appeared suddenly in the conclusion. We apologize for this mistake. Thus, in the revised version of the manuscript, we have changed "SNC" to "CNS".
Reviewer 2 Report
The work is interesting and well presented. In addition, the crested porcupine is included in a list of endangered species. The combination of traditional and imaging techniques gives this paper an additional value. However, there is a major concern that the authors should take into consideration. The levels indicated by the lines in Figure 1 do not correspond to the brain images shown in the figures. For example, line IV goes through the cranial thalamus. Thus, it is not possible with that line path to see either the cerebellum or the cerebellar peduncle in Figure 5. It is not possible to see the cerebellar peduncle before the cerebral peduncle, which is identified in the ventral part of Figure 6 (more caudal). Another example is line V, that goes through the mesencephalon. Thus, Figure 6 does not correspond to that level. Here the III ventricle is clearly seen, whereas line V crosses the mesencephalic aqueduct. It is practically the same in all images, which should be reviewed before a possible publication.
There are no comments on the quality of English Language. I consider that minor changes are needed.
Author Response
Dear Reviewer,
We concur entirely with your observation that the depicted levels indicated by the lines in Figure 1 do not align with the brain images presented within the figures. This discrepancy arose from the acquisition of anatomical sections in an oblique transverse plane, resulting in quite oblique transverse magnetic resonance (MR) images. In the revised edition of the manuscript, we have undertaken the task of reconstructing Figure 1 to establish a more accurate correlation between the lines and the brain images. Moreover, we have reordered some figures that were wrongly enumerated.
Round 2
Reviewer 1 Report
Dear authors,
Thank you very much for revising the manuscript, it is greatly improved.
I have some minor points to report.
- The title, methods, and results should have the same sequence (MRI followed by anatomical sections)
Replace gross sections by cross sections in figure legends too
Author Response
Dear Reviewer,
We sincerely appreciate your comments and suggestion because they were pretty helpful in improving the quality of our manuscript.
- As you recommend, we have changed the "title, methods, and results", which have now the same sequence (anatomical sections followed by MRI)
- We have replaced gross sections with cross sections in figure legends as you suggest
Reviewer 2 Report
The revised version is a great improvement compared to the previous one. All mistakes have been corrected, specially those related to conceptual anatomy. I find the paper very interesting and provides much information to all anatomist, particularly those interested in exotics. Thus, the paper fulfills the requeriments of Animals and can be published in its final version.
There are no comments. The paper has just minor errors.
Author Response
Dear Reviewer,
We appreciate your comments because they were pretty helpful in improving our manuscript.
Sincerely